# Patient Satisfaction with a Dedicated Infusion Pump for Subcutaneous Treprostinil to Treat Pulmonary Arterial Hypertension

**DOI:** 10.3390/jpm13030423

**Published:** 2023-02-26

**Authors:** Marcin Waligóra, Barbara Żuławinska, Michał Tomaszewski, Pere Roset, Grzegorz Kopeć

**Affiliations:** 1Pulmonary Circulation Centre, Department of Cardiac and Vascular Diseases, Faculty of Medicine, Jagiellonian University Medical College Centre for Rare Cardiovascular Diseases John Paul II Hospital in Krakow, 31-202 Krakow, Poland; 2Department of Cardiology, Medical University of Lublin, 20-090 Lublin, Poland; 3Ferrer International, 08029 Barcelona, Spain

**Keywords:** subcutaneous infusion pump, treprostinil, pulmonary arterial hypertension, patient satisfaction, nurse satisfaction

## Abstract

*Background and Objectives*: Parenteral prostacyclins are crucial in the pharmacological treatment of pulmonary arterial hypertension (PAH). Indeed, subcutaneous administration of treprostinil has been associated with considerable clinical and hemodynamic improvement, right-sided heart reverse remodeling, and long-term survival benefit. However, evidence on patient perceptions about handling a subcutaneous infusion pump for self-treatment administration and nurse views about training the patients are lacking. This study aimed to describe the perception of PAH patients and nurses regarding the use of the new portable I-Jet infusion pump for the self-administration of subcutaneous treprostinil, as well as its real-world training needs. *Materials and Methods*: The study is an open, observational, prospective, single-center, non-interventional study. Patients with PAH on stable therapy with subcutaneous treprostinil were invited to take part in the study at their start of use of the portable I-Jet infusion pump for the self-administration of treatment. Participants filled in a questionnaire to report their satisfaction with the use of the pump, as well as their compliance, confidence, convenience, preferences, technical issues, and perceptions of the training they received. *Results*: Thirteen patients completed the questionnaire after being on the pump for 2 months: 69% were females and the mean age was 51 years. The most frequent PAH etiologies were congenital heart disease (46.2%) and idiopathic PAH (38.4%). Most patients were either World Health Organization (WHO) functional class II (53.8%) or III (46.2%). Ten patients (76.9%) found the pump easy and convenient to live with. All patients declared themselves to be fully compliant and confident in using the pump (*n* = 13) at the end of the study follow-up. Ten patients (76.9%) would choose the new pump in the future. None of the patients made reference to technical issues that required additional hospital visits. Eight patients (61.6%) reported that learning how to use the I-Jet infusion pump was easy or very easy, and none considered that further training was needed. One trainer nurse was interviewed and confirmed the satisfaction of patients and the simplicity of usage and training. *Conclusions*: PAH patients were highly satisfied with the use of the new portable I-Jet infusion pump for self-administering subcutaneous treprostinil. Convenience and ease of use were valuable and commonly reported features. Moreover, the training requirement was simple. These preliminary findings support the routine use of the I-Jet infusion pump.

## 1. Introduction

Pulmonary arterial hypertension (PAH) is a rare form of pulmonary hypertension characterized by progressive obliterative vasculopathy of the distal pulmonary arterial circulation that usually leads to right ventricular failure and death [1,2]. Parenteral prostacyclins are a key part of the medical therapy for PAH [3,4]. Continuous administration of the drug, either subcutaneously or intravenously, is required [2]. Subcutaneous (SC) administration of treprostinil has been associated with considerable clinical and hemodynamic improvement, right side heart reverse remodeling, and increased long-term survival benefit [5,6]. SC administration using infusion pumps implies less risk of systemic infection or catheter dislocation than intravenous administration [7] and is sometimes considered to have less life-threatening complications [8,9] and is easier to use for patients having difficulties with intravenous therapy [10].

Continuous SC treprostinil infusion is delivered through an SC catheter, employing an ambulatory infusion pump (Figure 1). Patients must be thoroughly trained in self-use and programming of the pump as well as the connection and care of the infusion set. Several different infusion pumps have been used to administer treprostinil subcutaneously, most of them resembling SC insulin infusion pumps [11]. In Poland, three different subcutaneous infusion pumps are available, including the I-Jet infusion pump, the micro SC infusion pump (Apex, China), and the Canè Crono (Canè, Italy) (Appendix A). The new I-Jet infusion pump (Everaid^®^, South Korea) is an ambulatory pump that has been designed exclusively for SC administration of treprostinil and is approved in the European Union (EU) (European CE (Conformité Européene) mark by CE Mark Notified Body 1639; Ferrer is the legal representative) [12]. The design of the I-Jet infusion pump was oriented towards incorporating features that improve ease of use to benefit PAH patients’ comfort (Table 1) [12].

Patient perspectives and the route of administration in long-term chronic conditions can enhance adherence and persistence. Better treatment adherence and persistence, improved clinical and therapeutic outcomes, and better health-related quality of life might be several outcomes of I-Jet pump use [13,14,15]. Nevertheless, patients with chronic conditions report higher adherence and persistence with treatment, confident self-management, and lower total costs with nurse support, medication reminders, injection training, and pen disposal [16]. Moreover, nurse support improves patient self-management of treatment recommendations and enables patients to discuss their adherence problems with healthcare [17].

This manuscript describes the experience of PAH patients using the new portable I-Jet infusion pump. Satisfaction and reliability for first-time users are described as well as patients’ compliance, convenience, preferences, and perceptions of technical aspects of the pump and the training received in hospital. The trainer nurse’s views on patient satisfaction and training needs were assessed. 

## 2. Materials and Methods

### 2.1. Study Design

All participating patients had their pumps changed (from Apex to I-Jet) due to a substitution of infusion pumps by the hospital provider in mid-2021 as consequence of a public tender. Physicians had no role in the hospital public tender. This circumstance was an opportunity to assess how changing the device may affect the user’s life. We decided to conduct the observational study described in this manuscript. This was an open, observational, prospective, single-center, non-interventional study designed to describe the satisfaction of PAH patients and their trainer nurse with the portable I-Jet infusion pump after 2 months of use. The study was conducted at the Department of Cardiac and Vascular Diseases, John Paul II Hospital, Krakow, Poland. 

### 2.2. Study Objectives

The primary objective of the study was to assess whether the I-Jet infusion pump was a satisfactory and reliable self-management option for first-time users.

The secondary objectives of the study were to explore:
Satisfaction: determine overall satisfaction assessment, most-liked features, service satisfaction, and I-Jet infusion pump setup, design, size.Technical performance: evaluate technical performance of the I-Jet infusion pump and occlusion alarms. Benefit: determine convenience of living with the I-Jet infusion pump and patients’ acceptance of the system.Quality of life: determine quality of life improvement or not worsening Education: identify educational aspects for improving the use of the I-Jet infusion pump. Trainer’s insight: describe the nurse experience.

### 2.3. Population

All patients were adults (older than 18 years) with an established PAH diagnosis. Patients who were on stable therapy with subcutaneous treprostinil, with no PAH worsening signs in the previous 3 months, no change in PAH-specific treatment, were on stable doses of diuretics, and attended John Paul II Hospital, Krakow, Poland were consecutively recruited into the study between August and October 2021. Patients signed the informed consent form (ICF), were recruited into the study, and changed the previously used device (Apex pump) to the portable I-Jet infusion pump. The exclusion criteria were any serious disease that could contraindicate participation in the study according to the investigator’s judgement; any psychological and/or physical condition that may negatively affect the proper follow-up of study procedures, such as any serious, not corrected hearing and/or visual loss, or difficulties in self-managing the I-Jet infusion pump, including any physical, cognitive, or behavioral limitations; being under legal protection; and any other health ailment that would preclude self-care. All patients were treated according to the European Society of Cardiology clinical practice guidelines [2].

### 2.4. Research Tools

The Patient Satisfaction Questionnaire (PSQ) (Appendix A) was developed to assess the satisfaction of patients with the I-Jet infusion pump. The questionnaire included 30 items. Likert-type responses ranged from 1 (very dissatisfied), 2 (dissatisfied), 3 (neither dissatisfied nor satisfied), 4 (satisfied), to 5 (very satisfied). The questionnaire included 8 questions on pump features and technical performance; 4 questions on pump set up/usability; 7 questions on daily use/simplicity; 4 questions on training, learnability, and supplemental training materials; and 7 questions on additional services and recommendations for patients. 

The Patient Benefit and Education Questionnaire (PBEQ) (Appendix A) was developed to collect data about the convenience of the I-Jet infusion pump from the patient’s point of view and best practices and educational aspects. The questionnaire included 17 questions: 10 questions had a yes/no answers; 4 questions required the most convenient option to be chosen; and 3 questions had a Likert-type response scale.

The Cambridge Pulmonary Hypertension Outcome Review (CAMPHOR) questionnaire is an established, validated, PAH-oriented tool for the measurement of health-related quality of life (Appendix A). The questionnaire was used to evaluate the quality of life at baseline and at the end of the study for patients with the I-Jet infusion pump. The CAMPHOR questionnaire contained 65 items measuring symptoms (25 questions), activity (15 questions), and quality of life (25 questions). Symptoms and quality of life were both scored out of 25, and activity was scored out of 30. Scores were negatively weighted so that a higher score reflected worse quality of life and greater functional limitation [18]. 

Medical device incidences (MDIs) and adverse drug reactions (ADRs) were reported through the phone during the duration of the study. The study was managed by an independent clinical research organization (CRO) and included pharmacovigilance supervision on safety concerns.

### 2.5. Data Collection 

After screening (visit 0), the study consisted of 3 data collection visits: visits 1 and 3 were on site and visit 2 was a telephone call. At visit 1, the ICF was signed off; patients were trained on the use of the I-Jet infusion pump by a nurse; they took home basic information on the disease; the pump was provided; and clinical, demographic, and quality of life (CAMPHOR) data were collected. The PBEQ was provided and was to completed after 28 days of using the pump. After 1 month, patients received a follow-up telephone call to collect MDIs and ADRs, resolve technical issues, and assess the patient’s compliance and ability to self-manage the pump. A reminder to answer the PBEQ was given. Visit 3 (8 weeks) took place at the hospital and responses to the CAMPHOR questionnaire, PBEQ, and PSQ were collected. Moreover, pump functionality was checked and treatment was optimized if necessary (Figure 2). At the end of the study, the trainer nurse was invited to provide insights on the training experience in a 30 min, in-depth qualitative interview conducted by telephone (Appendix A). 

### 2.6. I-Jet Infusion Pump

The I-Jet infusion pump is a CE marked infusion pump suitable for continuous subcutaneous infusion of liquid medicine (Figure 3) [12]. It has an icon-based color screen and is equipped with alarms and vibrations for patient safety. The pump runs quietly and continuously in the background 24 h a day. Treprostinil is stored in a 3 mL syringe inside the pump, which needs to be changed every 3 days [12].

### 2.7. Training 

Patients were trained on the use of the I-Jet infusion pump by a trainer nurse with 20 years’ experience in training patients treated with subcutaneous treprostinil. Training focused on how to use the pump, frequency of changing the infusion site, actions to be taken in case of an occlusion alarm, and changing the treprostinil syringes (Appendix A). Training was conducted by the same instructor, a research nurse employed by John Paul II Hospital. 

### 2.8. Statistical Analysis 

Descriptive statistics were performed. For continuous variables, the number of patients, means, standard deviations, medians, and minimum and maximum values were determined. For categorical variables, absolute and relative frequencies were ascertained. 

### 2.9. Ethical Considerations

The study was conducted in accordance with the Helsinki Declaration, and all patients gave their consent to participate in the study. The local bioethical committee approved the protocol of the study (2KBL/OIL/2021).

## 3. Results

### 3.1. Baseline Characteristics

In early 2021, a total of 143 patients with an established diagnosis of PAH were treated at John Paul II Hospital, Krakow, Poland: 14 had reactive PAH and were, therefore, treated with calcium channel blockers instead of PAH targeted therapy; 18 of the remaining 129 patients were candidates to receive SC treprostinil. These 18 patients were identified, contacted, and informed of the planned change of the infusion pump. Before changing the pump, 3 of them died and 1 was switched from treprostinil to inhaled iloprost for medical reasons. Among the remaining 14 patients, 1 had a right heart decompensation and did not meet the inclusion criteria to entry the study, while 13 patients provided consent and participated in the survey. 

The mean age was 50.8 (standard deviation (SD): 14.4) years and 69.2% of participants were females (Table 2). The most frequent PAH etiologies were congenital heart disease (46.2%) and idiopathic PAH (38.4%). A total of 53.8% of patients were assessed as having World Health Organization (WHO) functional class II and 46.2% had WHO functional class III. The median time from PAH diagnosis was 7 years and from initiation of therapy with treprostinil was 13 months. All patients had at least one comorbidity and 76.9% had at least 3 different comorbid conditions. The most frequent comorbidities were chronic cardiac failure (92.3%), metabolism and nutrition (38.5%), and blood and lymphatic system (30.8%) disorders. The mean (SD) 6 min-walk distance (6MWD) was 380 (129) m at the first visit and 375 (131) m at the third visit.

Among the 13 participants, 6 (46.2%) were on SC treatment for 1 to 3 years and the remaining 7 (53.8%) were on SC therapy for at least 3 months. The median dose of treprostinil during the study was 39.5 (12.9) ng/kg/min with a stable flow rate across the 3 study visits [Mean = 3.876 mL/hour].

### 3.2. Patients’ Satisfaction

Overall, 76.9% of the patients were satisfied with the I-Jet infusion pump (Figure 4).

### 3.3. Patients’ Satisfaction with Physical Characteristics of the I-Jet Infusion Pump

Regarding the style and design and size and weight of the I-Jet infusion pump, 9 (69.2%) and 7 (53.8%) patients were satisfied, respectively. A total of 10 (76.9%), 10 (76.9%), and 11 (84.6%) patients were satisfied with the usability of the keyboard, access to the battery, and access to the syringe compartments, respectively. A total of 53.8% of patients were neither dissatisfied nor satisfied with the legibility of the LCD color display (Figure 5). 

### 3.4. Patients’ Satisfaction with Usability Characteristics of the I-Jet Infusion Pump

Here, 92.3% (12 out of 13) of the patients were satisfied with the fact that the pump was water resistant and that the flow rate could be set at very precise 0.001 mL steps. 

Patients were also satisfied with the easy access to the settings menu (*n* = 9, 69.2%); the user-friendly design of the software interface (*n* = 9, 69.2%); the content shown on the display (*n* = 7, 53.8%); the handling of the i-life syringe (*n* = 11, 84.6%); and the battery change process (*n* = 12, 92.3%) (Figure 6). 

### 3.5. Technical Performance of the I-Jet Infusion Pump

Regarding the technical performance of the I- JET infusion pump, only one patient reported an occlusion alarm at the third visit. None of the patients reported technical issues that would require a hospital visit over the 1-month period of observation. None of the patients reported errors that would lead to replacement of the I-Jet infusion pump. 

### 3.6. Patients’ Reported Benefits

All patients (100%) declared themselves to be fully compliant and confident with the use of the I-Jet infusion pump (*n* = 13) at the end of the 1-month study follow-up, and 70% declared that the I-Jet infusion pump did not hinder them from carrying out any everyday activities in their lives. A total of 11 (84.6%) patients reported good control of SC administration of treprostinil with the I-Jet infusion pump. Among the 13 patients, 69.2% changed their infusion site every 3–4 weeks. The arm was the preferred infusion site. Out of 13 patients, 10 (76.9%) found the device easy/convenient to live with and 76.9% preferred to use the I-jet infusion pump in the future (Figure 7).

### 3.7. Health-Related Quality of Life

No changes in the scores of symptoms, activities, or quality of life were observed at the end of the study compared to baseline (Table 3). The CAMPHOR scores indicated that patients perceived good quality of life with no physical limitations or symptoms throughout the study.

### 3.8. Education and Training 

According to the responses, 9 (69.2%) patients were satisfied with the quality of the instructions, the training materials, and the overall I-Jet infusion pump service, while 10 (76.9%) patients were satisfied with the comprehensibility of the operating instruction manual (Figure 8). 

All patients (*n* = 13, 100%) considered that the training for first use of the I-Jet infusion pump was appropriate. For 8 (61.6%) of the patients, learning to use the I-Jet infusion pump was easy, and all agreed that a re-training session was not necessary (Figure 9).

### 3.9. Safety Evaluation

All patients (100%) reported that administration with the new I-Jet infusion pump was safe. No MDIs or ADRs were reported during the duration of the study.

### 3.10. Nurse Insight

The trainer nurse was interviewed at the end of the study. She had been practicing PAH nursing for 20 years. She was a full-time employee at the pulmonary hypertension center at John Paul II Hospital. She had experience in managing CADD MS3^®^ (Smiths Medical) and micro SC (Apex^®^) infusion pumps and this was her first time using the I-Jet infusion pump. 

The nurse reported that all patients only needed support from a healthcare professional on the infusion pump features and functionality at the beginning of therapy. She confirmed the ease of training and satisfaction of patients with the I-Jet infusion pump. She was very satisfied with the I-Jet infusion pump’s size, display, and refill process, the keypad, the alarm set up, and the occlusion alarm. She agreed that the I-Jet infusion pump was a reliable self-management device for PAH patients. She felt that the I-Jet infusion pump may have improved patients’ quality of life. 

## 4. Discussion

This study described the real-life experience and satisfaction of PAH patients after switching to the new portable I-Jet infusion pump for the first time and using it for two months. Patients were highly satisfied with its physical aspects, such as the design, size, and weight, as well as the technical features of the pump. Most PAH patients found its use comfortable and safe. Simplicity of handling and comfort for carrying out everyday life activities were highly rated. According to other research findings, the portability and small size of the infusion pump were related to the satisfaction of patients [19]. Compared with the use of a traditional fixed-syringe pump, the I-Jet infusion pump was easy to use and allowed freedom of movement and comfort in performing common activities of daily living, which increased the satisfaction of patients and nurses alike [19]. Better disease control and reduced time spent on drug preparation are other features of infusion pumps that are considered of importance for patients [20,21].

Patient satisfaction with treatment is highly relevant in the management of PAH for which there are multiple approved therapies available [22]. However, meeting patients’ needs and expectations might be challenging due to various type of drugs and their administration, with each switch hampering QoL. Changes in effectiveness scores have been correlated with changes in 6MWD scores and effectiveness, convenience, and satisfaction scores have been significantly associated with improvements in PAH-specific QoL [23]. Likewise, the use of auto-injection devices for administering SC methotrexate in rheumatoid arthritis patients increased treatment adherence due to improved patients’ preference and satisfaction [16]. In this study, high satisfaction with the I-Jet infusion pump coexisted with a high percentage of PAH patients reporting that they were confidently managing SC administration of treprostinil and fully compliant with the treatment after 2 months. Additionally, participants maintained an excellent quality of life throughout the study period.

Patients’ adherence to treatment is a key factor for improving chronic disease control (including destabilization of one clinical condition), efficient use of healthcare resources, and for reducing disease costs [24,25]. Patients’ attitudes and adherence towards their medicines are influenced by many factors, including their perceived (or real) benefits and drawbacks, previous experience of use, perceptions of their illness, satisfaction with treatment, and personal preferences [26]. In our study, the experience of PAH patients with the I-Jet infusion pump use was highly satisfactory, implying that a high level of adherence to therapy can be expected.

Being able to perform daily activities is of high importance for chronically ill PAH patients and may have a favorable effect on their quality of life and mental health [27]. Stable PAH patients should be encouraged to be active. It has been demonstrated that patients who keep performing their daily activities or increase their level of physical activity improve their distance in 6MWD compared to sedentary patients [28]. The I-Jet infusion pump helped PAH patients to comfortably carry out their daily activities.

Compared to other pumps used for the SC administration of treprostinil, common malfunctions were reported with the MiniMed^®^ Model 506 (Medtronic) infusion system (55 patients in the treprostinil group (24%) and 77 patients in the placebo group (33%) [6]. Likewise, Barst et al. (2006) described the results of an open label, SC treprostinil long-term extension study over 4 years in 860 patients [29]. Delivery system complications were reported by 255 patients (30%), with the most frequent being pump malfunction (222 patients; 26%) and problems with the infusion set (74 patients; 9%) [29]. In contrast, patients using I-Jet infusion pump in this study reported no MDRs, ADRs, or malfunctions of the device at two months of prospective observation.

During this study, a trainer nurse trained PAH patients to use the pump at the beginning of the study, with no need for further training or monitoring over time. The trainer nurse confirmed the results obtained from patients in terms of satisfaction, compliance, comfort, and confidence. Ease of training and learning to use a device is associated with increased adherence and adequate control of therapy administration [30] and more efficient use of nursing time [19]. 

This study had limitations and its findings need to be interpreted accordingly. The absence of a comparator prevented us from testing the robustness and true magnitude of the results obtained with the I-Jet infusion pump. Its open, descriptive nature implied low internal validity; however, external validity was high as the findings reflected everyday life. The study was conducted in a small sample of patients in a single center over a short follow-up period due to low prevalence of PAH and infrequent use of SC treprostinil to treat PAH. Additionally, in Poland, all PAH patients have their available pharmacological treatments fully reimbursed and accessible. In consequence, treatment with SC treprostinil depends solely upon medical decision-making and patient agreement in minimizing potential selection bias. Self-reported information on a questionnaire may be biased by patients’ interpretation of questions and answers. The questionnaire was not validated for the assessment of patient satisfaction and content bias may arise in consequence. Despite these limitations, this study presents preliminary findings on the experience and satisfaction of PAH patients treated with SC treprostinil with the I-Jet infusion pump.

The study provides evidence of PAH patients’ satisfaction with treatment delivery devices in real life. The adequate, easy, and safe self-management of SC treprostinil with the I-Jet infusion pump may reduce hospital visits and improve patients’ quality of life with no additional physical limitations in daily life. Finally, the study also provides information about patients’ training needs for the design of new infusion pumps.

## 5. Conclusions

PAH patients were highly satisfied with the I-Jet infusion pump for the self-administration of SC treprostinil. Convenience, comfort, safety, and simplicity were features of the infusion pump that were highly valued by patient users. Its straightforward instructions and undemanding training were also prized. The trainer nurse’s assessment aligned with patients’ perceptions and high usability satisfaction.

## Figures and Tables

**Figure 1 jpm-13-00423-f001:**
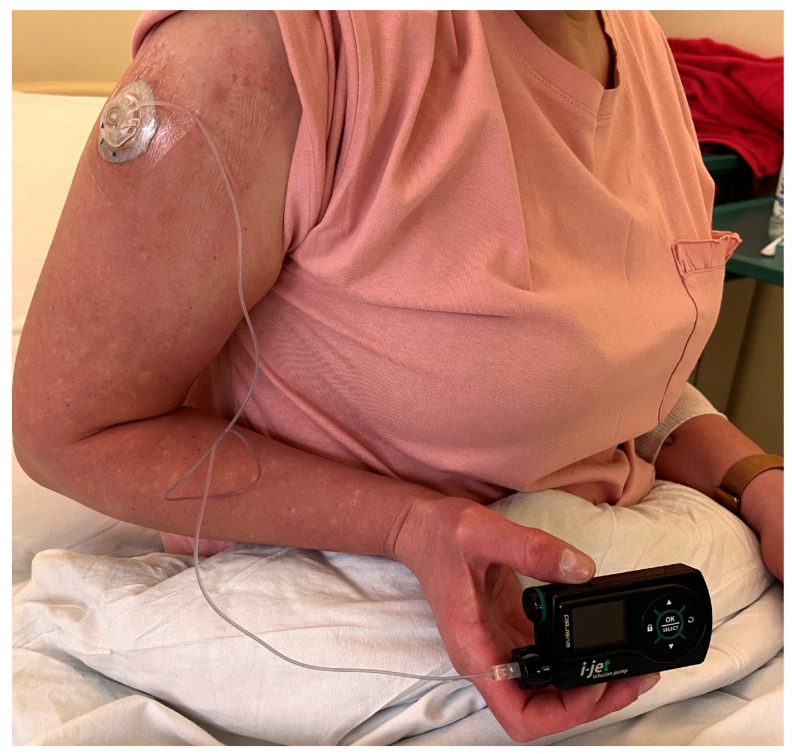
Patient with the I-Jet infusion pump.

**Figure 2 jpm-13-00423-f002:**
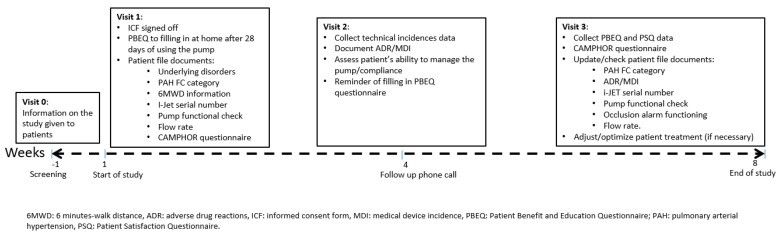
Study schedule.

**Figure 3 jpm-13-00423-f003:**
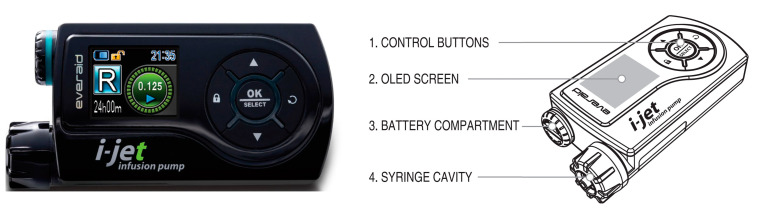
I-Jet infusion pump.

**Figure 4 jpm-13-00423-f004:**
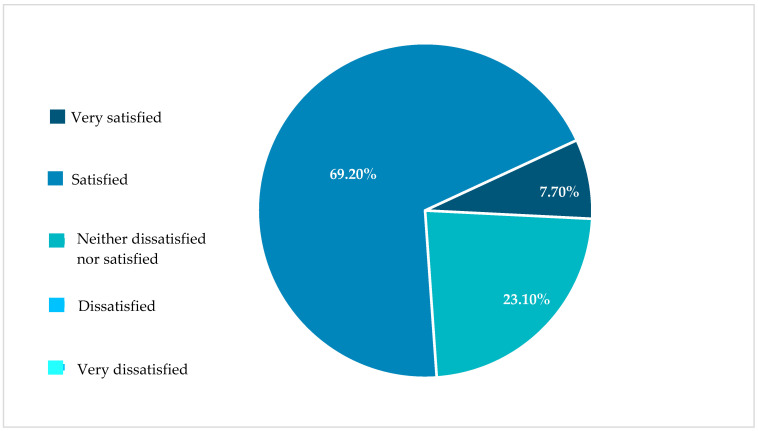
Patients’ overall satisfaction with the I-Jet pump in percentages.

**Figure 5 jpm-13-00423-f005:**
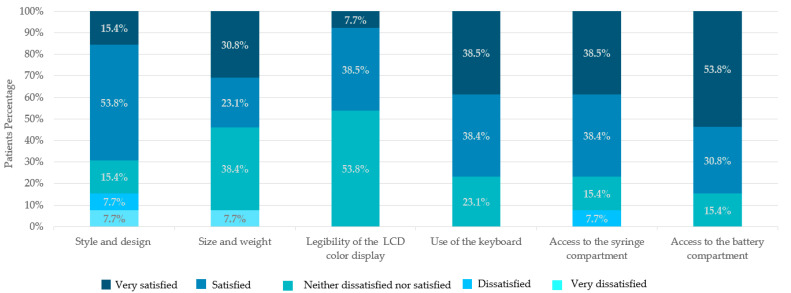
Patients’ satisfaction with physical features of the I-Jet pump.

**Figure 6 jpm-13-00423-f006:**
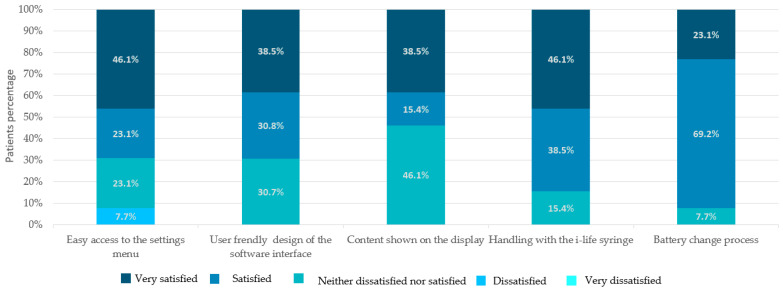
Patients’ satisfaction with usability characteristics of the I-Jet pump.

**Figure 7 jpm-13-00423-f007:**
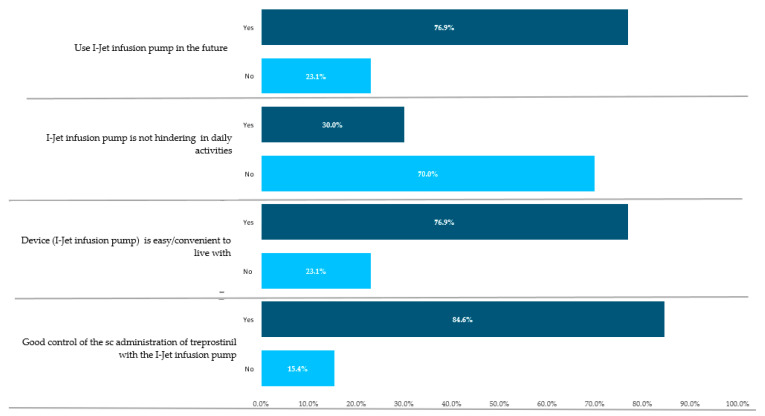
Patients’ benefits of the use of I-Jet infusion pump.

**Figure 8 jpm-13-00423-f008:**
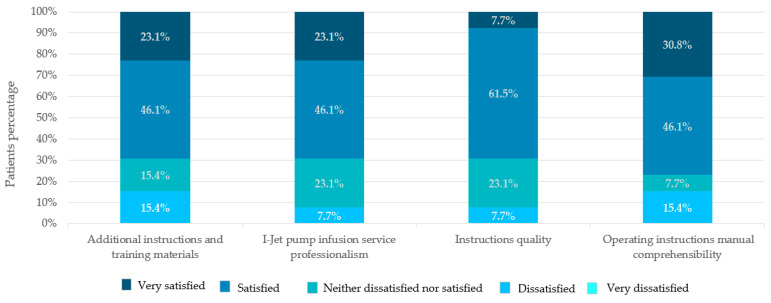
Patients’ satisfaction with I-Jet infusion pump education.

**Figure 9 jpm-13-00423-f009:**
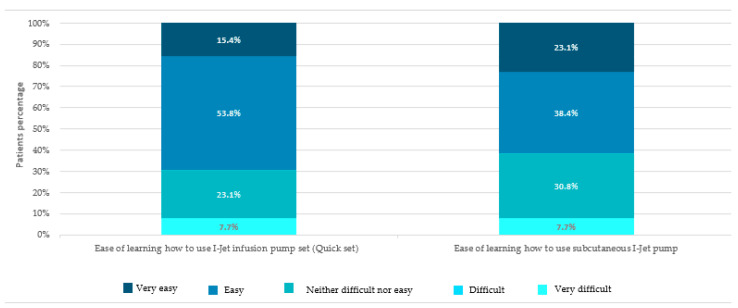
Patients’ perception of the learning process for using the I-Jet infusion pump.

**Table 1 jpm-13-00423-t001:** I-Jet pump features and expected benefits for PAH patients.

Features	Benefit
Capable of adjusting small increments of infusion rates (0.001 mL/h) as prescribed with ±5% accuracy	Light weight and prostacyclin-sensitive patients can be more precisely titrated, decreasing the risk of adverse events from inappropriate infusion rates
Warning message and alarm sounds every 10 min if infusion is stopped	Reassurance about being notified if infusion stops Infusion can be restarted at any time if necessary
No bolus function	No risk of accidental bolus infusion
Self-test function	Reassurance about the pump full functionality
Standard Luer Lock connector	Use with standard lines and needles
Dual Action Pump Lock Mode	Prevents accidental unlocking
Vibration alert	Reassurance about safe function

**Table 2 jpm-13-00423-t002:** Demographic and clinical characteristics of patients at baseline.

Characteristic	Patients (N = 13)
Age (years) (mean, SD)	50.8 (14.4)
Gender (*n*, %)	
Male	4 (30.8)
Female	9 (69.2)
Body weight (kg) (mean, SD)	70.3 (14.6)
PAH etiology (*n*, %)	
Connective tissue disease	1 (7.7)
Congenital heart disease	7 (53.8)
Idiopathic PAH	5 (38.4)
WHO Functional class (*n*, %)	
Class I	0 (0.0)
Class II	7 (53.8)
Class III	6 (46.2)
Class IV	0 (0.0)
Time from PAH diagnosis (years) (mean, SD)	8.5 (4.8)
Time from treprostinil therapy initiation (months) (mean, SD)	13.0 (3.5)
Patients with at least one comorbidity other than PAH (*n*, %)	13 (100)
Number of comorbidities (*n*, %)	
1	2 (15.4)
2	1 (7.7)
≥3	10 (76.9)
Comorbidities (*n*, %)	

Metabolism and nutrition disorders	5 (38.5)
Blood and lymphatic system disorders	4 (30.8)
Respiratory, thoracic and mediastinal disorders	4 (30.8)
Vascular disorders	4 (30.8)
Congenital, familiar and genetic disorders	3 (23.1)
Musculoskeletal and connective tissue disorders	3 (23.1)
Endocrine Disorders	2 (15.4)
Eye disorders	1 (7.7)
Gastrointestinal disorders	1 (7.7)
Immune system disorders	1 (7.7)
Neoplasms benign, malignant and unspecified	1 (7.7)
Renal and urinary disorders	1 (7.7)
Pulmonary arterial hypertension specific treatment	
Treprostinil	1 (7.7%)
Treprostinil and sildenafil	2 (15.4%)
Treprostinil with bosentan and sildenafil	9 (69.2%)
Treprostinil with macitentan and sildenafil	1 (7.7%)

N = number, kg: kilogram, PAH: pulmonary arterial hypertension, SD: Standard deviation, WHO: World Health Organization.

**Table 3 jpm-13-00423-t003:** Cambridge Pulmonary Hypertension Outcome Review (CAMPHOR) questionnaire results.

	Baseline Assessment	Follow-Up Assessment	*p*
Symptoms	6 (5–15)	7 (4–16)	0.48
Activities	11 (9–15)	12 (10–14)	0.11
Quality of life	5 (4–8)	6 (4–8)	0.93
Total	23 (18–39)	22 (18–43)	0.89

## Data Availability

Not applicable.

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
