# Peer review of "Patient Satisfaction with a Dedicated Infusion Pump for Subcutaneous Treprostinil to Treat Pulmonary Arterial Hypertension"

_jpm, 2023, doi:10.3390/jpm13030423_

Round 1
Reviewer 1 Report
The study reports on the perceptions of patients and a nurse regarding the use of the new portable I-Jet infusion pump for the self-administration of subcutaneous treprostinil in the treatment of pulmonary arterial hypertension (PAH). The study is as an open, observational, prospective, single center, non-interventional study. The results of the study suggest that PAH patients are highly satisfied with the use of the new I-Jet infusion pump, with most finding it easy and convenient to use and all reporting full compliance and confidence in using the pump.
However, there are a few suggestions for improvement that the authors should consider:
1) Comparison table: The authors can include a table providing a comparison of the available infusion pumps used for the subcutaneous infusion of treprostolil including size, weight, battery life, minimally adjustable drug delivery rate, delivery accuracy and other characteristics. This would provide a clearer picture of the benefits of the I-Jet infusion pump.
2) Approval status: The authors should clarify whether the I-Jet infusion pump is approved by the European Medical Agency (EMA) or only in France.
3) Reason for change: The authors mention that the infusion pumps were changed due to the company delivering the infusion pumps deciding to change to the I-Jet infusion pump. The authors should explain the reason for this change and provide for details.
4) Do the PAH patients have other options of infusion pumps in Poland, or currently this one is only available one?
5) Old vs new comparison: As the patients had experience with using two different infusion pumps, the authors should consider asking the same questions about the differences in the use of the old vs new I-Jet infusion pumps to gain a clearer understanding of patients' perceptions of the two pumps.
6) The authors can consider providing an example picture of a patient with the I-jet infusion pump, so that the reader can have an idea what it would look on a patient (an anonymous picture without providing patients details and face with the agreement).
In conclusion, the study provides valuable insights into patients' and nurses' perceptions of the I-Jet infusion pump for the treatment of PAH. The suggestions provided can further strengthen the study and provide a more comprehensive understanding of the use of the pump.
Author Response
Manuscript “Patient satisfaction with a dedicated infusion pump for subcutaneous treprostinil to treat pulmonary arterial hypertension”
Dear reviewers and editors,
Thank you for reviewing our manuscript and for giving us valuable feedback on it.
The manuscript has been revised and your comments considered in the way we describe in attachment.
We hope that now the manuscript fulfills the required standards to be considered for publication in the Journal of Personalized Medicine.
Looking forward to receiving your comments.
Thank you.
Kind regards,
The co-authors.

Reviewer 2 Report
The authors described the patient satisfaction with a dedicated infusion pump for Pulmonary arterial hypertension (PAH), which is a rare form of pulmonary hypertension. The manuscript is well written based on a prospective cohort study in a single center in Poland and can have potentially important implications in future practice
Here are some comments to help improve the manuscript:
1) I am a bit surprised about the sample size of being only 13. Can you please explain a bit on why the sample size is so limited and whether there is any criteria of precluding more patients being included or this infusion pump itself has low acceptance rate?
2) Because of the limited sample size, do you think there is any selection bias that could be due to health conditions, social economic status, access to health care insurance, etc. that might impair the generalizability of your findings?
3) Can you please discuss the implication of your study in the discussion section?
4) In the limitation section, I think the author should also mention the limited sample size, and the single center intrinsic limitations.
Author Response

(The authors gave the same response as above.)

Round 2
Reviewer 1 Report
I thank authors for their considerations of my previous comments. I have no more comments. All the best.